# Comparison of Negative Pressure Wound Therapy (NPWT) and Classical Wet to Moist Dressing (WtM) in the Treatment of Complicated Extremity Wounds in Children

**DOI:** 10.3390/children10020298

**Published:** 2023-02-03

**Authors:** Milan Slavkovic, Dragoljub Zivanovic, Siniša Dučić, Valentina Lasić, Nado Bukvić, Harry Nikolić, Vlatka Martinović

**Affiliations:** 1University Children’s Hospital, 11000 Belgrade, Serbia; 2Faculty of Medicine, University of Niš, 18000 Niš, Serbia; 3University Clinical Center Nis, Clinic for Pediatric Surgery, Orthopedics and Traumatology, 18000 Niš, Serbia; 4Faculty of Medicine, University of Belgrade, 11000 Belgrade, Serbia; 5Department of Pediatric Surgery Clinic for Surgery, University Clinical Hospital of Mostar, 88000 Mostar, Bosnia and Herzegovina; 6Faculty of Medicine, University of Mostar, 88000 Mostar, Bosnia and Herzegovina; 7Clinical Hospital Center Rijeka, Department for Pediatric Surgery, 51000 Rijeka, Croatia; 8Faculty of Medicine, University of Rijeka, 51000 Rijeka, Croatia

**Keywords:** negative pressure wound therapy, wet to moist dressing, complicated wound, pediatric population

## Abstract

Treating complicated wounds in the pediatric population using traditional wet to moist wound dressing methods is not always appropriate due to the frequent need to change dressings daily or even a number of times a day, causing distress to the patient. Topical negative pressure is a method that allows for fewer dressings and provides localized benefits, thus accelerating wound healing. The merits of this therapy have been proven in studies on adults, but research on the pediatric population is scarce. Here we intend to present the results of negative pressure wound therapy (NPWT) on 34 pediatric patients (study group) and compare them with 24 patients (control group) treated with the traditional wet to moist dressing for complicated wounds. The results show that topical negative pressure wound therapy is a safe method that downgrades a wound from a complicated to a simple one and allows definitive coverage using a simpler technique with fewer wound dressings. The scars of the patients in the study group exhibited a better result on a visual scar scale. The patients in the control group had a shorter hospital stay. Based on the recorded results, we were able to make treatment recommendations.

## 1. Introduction

A “complicated wound” can have different definitions in the literature, but the term mainly refers to a combination of a large tissue defect with an active or threatening infection [1]. These are usually chronic deep or full-thickness wounds that heal slowly or do not heal at all. Such wounds are contaminated, regardless of the cause, size, location, and method of treatment; whether or not an infection will occur depends on the virulence of the microorganism, the number and type of microorganism, as well as on local tissue perfusion and the patient’s immune competence [2]. The cause of such a wound can be trauma, a condition after an infection, or complex surgical intervention. Wounds may initially be uncomplicated, and later turn into complicated ones due to a large defect, inadequate surgical closure, ischemia, poor anastomosis, and infection. The presence of foreign material in the wound interferes with the healing process and increases the risk of infection and necrosis. The presence of infection in the wound is a significant aggravating factor because infected wounds, whether surgical or traumatic, heal slowly.

The wounds of pediatric patients can be complicated by skin defects and/or soft tissue, exposure of bone or neurovascular structures, and infection [3,4,5]. Traditional treatments requiring frequent wound dressings and repeated debridement are not well tolerated by children [6,7]. Therefore, daily dressing changes (or even several times a day) under general anesthesia or at least conscious sedation are necessary. Applying topical negative pressure in treating such wounds facilitates healing, decreases the time to definitive wound coverage, reduces the number and frequency of dressings, downstages reconstructive procedures, and allows earlier return to activities of everyday life [1,3,8,9,10]. Negative pressure wound therapy (NPWT) reduces local edema, improves vascular and lymphatic flow, diminishes bacterial contamination, and promotes angiogenesis [1,6]. Moreover, negative pressure produces microdeformations on the cellular and tissue level, resulting in cell proliferation and migration [11]. The effects of negative pressure at both macroscopic and microscopic levels facilitate the formation of healthy granulation tissue [7], promoting spontaneous wound closure or closure with direct suturing or coverage with free split-thickness skin grafts (STSG) or local flaps [12]. Recently, the use of skin grafts has been combined with dermal regeneration templates [13,14]. Although numerous papers on the use of NPWT in children have been published, the FDA considers the evidence of efficacy and safety to remain insufficient in the pediatric population [15]. Moreover, there are only several papers that directly compare traditional wet to moist dressing and NPWT in the treatment of complicated extremity wounds in children.

This study aims to compare these two treatment options for complicated extremity wounds in children.

## 2. Materials and Methods

Herein we present a prospective single-institution study. Patients of pediatric age (<18 years) treated for complicated extremity wounds between February 2018 and December 2020 were included in this study. This study was conducted in accordance with the Declaration of Helsinki and approved by the Ethics Committee of University of Nis Faculty of Medicine (protocol code 12-1250/10, date of approval 06 February 2018).

The choice of treatment was the responsibility of the treating physician and based solely on their preferences and decision. Some patients were initially treated with classical dressing and then transferred to NPWT. Those patients were assigned to the NPWT group.

According to the treatment modality, patients were divided into the NPWT group (study group) and wet to moist group—WtM (control group). Further, both groups were divided into two subgroups. The first subgroup consisted of traumatic wounds (subgroup T) and the second (subgroup S) had surgically created wounds after excision and debridement of infection or another form of pathology.

Inclusion criteria were patients younger than 18 years of both sexes with complicated extremity wounds treated with NPWT or WtM. Exclusion criteria were complicated wounds outside extremity limits, uncomplicated wound, wounds treated with methods other than NPWT and WtM, and patients lost for follow-up.

The age and sex of the patients, mechanism of injury, indication for NPWT, time from injury to instituting treatment and from the beginning of treatment to instituting NPWT, duration of the treatment, number of wound dressings, duration of antibiotic and analgesic therapy, type of definitive wound closure, complications, duration of follow-up, and number of follow-up visits were recorded. Furthermore, the results of hematology, biochemistry and microbiology, as well as the number and quantity of blood transfusions were monitored. Exposure of bone, osteosynthesis, tendons, and neurovascular structures were also noted. The severity of the wound was assessed by calculating the wound score (WS) [16]. 

The wound area was calculated from photographs using the ImageJ ver. 1.53a program, after calibration.

All patients who received NPWT were treated using a V.A.C.^®^ therapy system device (Kinetic Concepts, Inc. (KCI), San Antonio, TX, USA); the models Info V.A.C or Acti V.A.C. VAC^®^ GranuFoam^®^ Dressing (KCI, San Antonio, TX, USA) were applied in all cases. The system was set to a continuous negative pressure of −100 to −125 mm Hg according to the patient’s age. After treatment of higher-priority injuries, thorough debridement was performed in the operating room (OR) under general anesthesia. Swabs for culture were also obtained. GranuFoam^®^ sponges were tailored to the wound shape, placed in the wound cavity, and covered with a plastic adhesive film. The amount and quality of evacuated fluid in suction canisters were closely monitored. Vital signs were recorded, and hematology and biochemistry tests were performed to monitor potential blood or protein loss. Dressing changes were scheduled in 3–5 day-intervals and performed under general anesthesia in the OR. We avoided scheduling these procedures during weekends, but it was performed immediately if an unscheduled dressing change was necessary. 

Definitive wound closure was performed with direct sutures, split-thickness skin grafts (STSG), local flaps, or a combination of these techniques. After fully integrating the grafts, the patients were discharged home or transferred to physiotherapy. Subsequently, follow-up visits were scheduled in the outpatient clinic. Scarring was assessed using the Observer Scar Assessment Scale (OSAS) [16]. 

Statistical analysis was performed by a specialist in medical statistics. Sample size calculation was performed using the G*Power statistical package. The SPSS statistical package (version 20) was used for statistical data processing. Data entry, tabular and graphical presentation were performed using the MS Office Excel program. The obtained results are presented tabularly and graphically with a textual comment. The normality of the distribution was tested with the Kolmogorov–Smirnov test and the Shapiro–Wilk test. Comparison of arithmetic means of two samples was performed by *t*-test, while in cases of irregular distribution of data, the non-parametric Mann–Whitney U test and Wilcoxon test for repeated measurements were used. The χ^2^ test was used to test the statistical significance of absolute frequency differences between samples. Determining the interdependence among the investigated variables was performed using the Pearson coefficient and the Spearman rank correlation coefficient. The statistical hypothesis was tested at the level of significance for the risk of α = 0.05, i.e., the difference between samples is considered significant if *p* < 0.05.

## 3. Results

In the study period, 58 patients met the inclusion criteria and were included in this study. Thirty-four patients treated with the V.A.C.^®^ therapy system were assigned to the NPWT group, while the 24 patients treated with classic wet to moist dressing constituted the WtM Group. 

Patient demographics and wound characteristics are summarized in Table 1. There were significantly more boys (29) than girls (5) in the NPWT group of patients. The mean age of the patients in both treatment groups was 13.00 ± 4.01 years. Patients in the NPWT group were significantly older (14.58 ± 2.85 years, range 6–18 years) than patients in the WtM group (10.83 ± 4.41 years, range 1–17 years), *p* < 0.001. There was no significant difference in the wound area between the two groups, although the wounds in the NPWT group (mean: 83.17 cm^2^; range 8.75–288.77) were slightly larger than those in the WtM group (mean: 58.38 cm^2^; range 4.2–240.90). The structures exposed in the wounds are listed in Table 1. 

Only the frequency of exposed osteosynthesis was significantly higher in the NPWT group (*p* < 0.05). Mixed Gram+ and Gram− isolates were most common in the NPWT group, while Gram+ isolates were predominant in the WtM group (Table 1). The wound score was significantly higher (*p* < 0.05) in the WtM group, indicating that the wounds were more complex in the NPWT group.

The main causes of traumatic wounds in both groups were traffic accidents, followed by injuries with agricultural machinery in the NPWT group and injuries from sharp or pointed objects in the WtM group (Table 2) and the main cause of wounds in the surgical excisional wounds group was soft tissue infection (Table 2).

Details of treatments are given in Table 3 and Table 4. The mean length of hospital stay of 36.44 days in the NPWT group was significantly longer compared to 19.46 days in the WtM group (*p* = 0.002). A period of 10.27 days on average from the injury to the beginning of NPWT may contribute to that difference.

The time from injury to definitive wound closure was also longer in the NPWT group (19.76 days) than in the WtM group (12.20 days) (*p* = 0.023). The number of dressing changes was significantly lower in the NPWT group (2.83 vs. 9.00; *p* = 0,009). The number of dressing changes in the outpatient clinic after discharge from hospital was significantly higher in the WtM group (0.97 vs. 2.54, *p* = 0001). Patients treated with classic wet to moist treatment experienced more days of fever (4.04) than patients treated with the V.A.C.^®^ therapy system (2.71), but the difference was insignificant. Intravenous antibiotics were administered on an empirical basis in patients from both treatment groups and then modified according to culture and antibiogram results, in consultation with the clinical pharmacologist. The decision to discontinue antibiotics or switch to oral was made after normalizing clinical and laboratory findings. The overall duration of antibiotic therapy was longer in the NPWT group (29.19 ± 24.36 vs. 14.50 ± 9.36 days; *p* < 0.001), but when the duration of antibiotic therapy after commencing NPWT was compared to the duration of therapy in the WtM group, the difference was insignificant (Table 3). 

When we compared the hematological and biochemical results between treatment groups, only statistically lower hemoglobin levels in the WtM group were observed (124.97 ± 17.18 vs. 93.30 ± 45.55 g/L; *p* = 0.011). We also observed a significant decrease in the WBC count and CRP levels during NPWT treatment compared to initial pretreatment levels (13.03 ± 5.02 vs. 10.09 ± 2.78 × 10^9^/L; *p* = 0.014 for WBC and 60.82 ± 46.61 vs. 25.32 ± 10.55 g/L; *p* = 0.009 for CRP). Patients treated with NPWT received a significantly greater amount of transfusion of erythrocytes compared to patients treated with the WtM dressing, while the difference in the amount of administered fresh frozen plasma was insignificant (Table 3). Patients in the NPWT group received about a third of the total amount of erythrocyte transfusion in the initial resuscitation phase of treatment before the initiation of the V.A.C.^®^ therapy system, indicating the severity of their injuries. Another third was received during V.A.C^®^ treatment, and the last third was received after definitive wound closure.

Minor complications (8) were noted only in the NPWT group and were solely related to the V.A.C^®^ therapy system. These minor complications were the obstruction of the drainage system (mainly Granufoam^®^ sponge, loss of vacuum, or drainage canister overfill). All these complications were addressed immediately, within an hour, as recommended. Modalities of definitive wound closure are presented in Table 4 and Figure 1, Figure 2, Figure 3 and Figure 4.

## 4. Discussion

NPWT has been used for many years in treating complicated wounds. Its efficiency and safety have been documented in numerous research papers, but mostly on adult patients. Even though NPWT has been used on the pediatric population, scientific papers documenting the results in this age group are limited to retrospective studies on a relatively small number of patients. There is only one paper that compares NPWT with the traditional wet to moist dressings in the pediatric population, and by the admission of the authors themselves, the research had too few patients to make a statistically significant conclusion [17]. The significance of our study is that it is the first prospective research with a statistically significant number of patients that compares NPWT and the traditional wet to moist wound dressing.

The mean age of the patients was 13.00 ± 4.01 years (range 1–18) with a statistically significant difference between the NPWT (14.58 ± 2,85) and WtM (10.83 ± 4.41) group. We believe that this difference affects the final result given that other researchers have found a correlation between patient age and wound severity [17], and it has also been demonstrated that in older pediatric patients, inflammatory cells appear later during the wound healing phase, which may negatively impact the outcome [18]. 

In addition to skin and soft tissue defects and wound infection, the presence of fractures in the wound as well as the exposure of bones, joints, tendons, neurovascular structures, or osteosynthesis makes the wound complex and significantly complicates the healing process, especially in children. A statistically significant difference in the exposure of structures between the group of subjects and the control group in our research existed only in the exposure of osteosynthesis, which was more often exposed in the group of subjects (8:1; χ2 = 4.024; *p* = 0.045). Removal of the osteosynthesis was necessary in only one patient. While in one patient the osteosynthesis was preserved with an external fixator, along with the rearrangement and replacement of part of the nails without affecting the position of the fragments and the healing of the fracture. The possibility of saving the exposed and infected osteosynthesis using NPWT is very significant in the treatment of both fractures and reconstructive procedures of bones and represents a major advantage of this method. This is supported by the results of Rentee et al., who reported a percentage of successful osteosynthesis rescue of 83% (in 10 out of 12 patients) after wound infection in patients treated with NPWT [19,20]. In all patients with bone, fracture, osteosynthesis and tendon exposure, the Granufoam^®^ sponge was applied directly to the exposed tissue/osteosynthesis. Only the exposed neurovascular structures were protected with a layer of Vaseline gauze before applying the Granufoam^®^ sponge.

The presence of infection makes the wound complicated, negatively affecting the length of treatment and often the quality of the scar. Given the deep infection around the osteosynthesis, the hardware sometimes needs to be removed. A traumatic wound can be primarily infected at the time of its creation or secondarily, during treatment. Surgically created wounds are often the result of surgical excision of infected soft tissue. Infection during the application of NPWT most often occurs in the area of the sponge that sometimes remains in the wound, which can mostly occur in large wounds where there are many blind pockets. In these patients, recovery is often longer and more interventions are necessary for the wound to heal. However, when used correctly, NPWT can reduce the number of bacterial species in the wound as well as the number of colonies in the wound [19]. To achieve this, a detailed treatment of the wound is necessary, both initially and with each dressing, which implies the complete removal of devitalized and necrotic tissue and purulent content of the wound. Negative pressure therapy is in no way a substitute for surgical wound debridement.

In all patients in this study, a wound swab was taken before the start of surgical therapy. Additionally, in patients with infected wounds on NPWT, a wound swab was taken every time the Granufoam^®^ sponge was changed. Statistically significantly more mixed (Gram+ and Gram−) microflora was isolated in the study group. Gram-positive microflora was most often isolated in the control group. In addition, there was no statistically significant difference in the duration of antibiotic therapy between the study and control groups. The duration of elevated body temperature was also not significantly different between the two investigated groups. Comparing laboratory values before and during NPWT in the study group shows a significant decrease in the values of leukocytes and CRP, i.e., inflammation parameters. Although infection is cited as one of the most significant complications of NPWT, the results of this study show that the application of negative pressure effectively suppresses infection in the treatment of complicated wounds.

Other authors have had similar experiences. In the study by McCord and colleagues of 82 wounds, 26 wounds had evidence of infection before initiation of NPWT, as evidenced by blood cultures, wound cultures, and frank pus in the wounds. NPWT was started following debridement and appropriate antibiotic treatment, and in the vast majority (22/26 or 88%), including two pressure ulcers with osteomyelitis, infection did not progress, and the wounds decreased in size and were ultimately closed [20]. 

Scoring systems that have also been recommended for evaluating acute traumatic wounds and open fractures, such as the ASEPSIS score and the National Nosocomial Infection Surveillance System score [21], are aimed primarily at predicting and evaluating surgical wound infection. For this reason, we used the wound score (WS) [16] in our study, which we considered to be the most comprehensive. A significantly lower value of the WS in the study group compared to the control group (*p* = 0.033) means that these wounds were more complicated and difficult to treat. The severity of wounds in the study group was significantly influenced by bone exposure, osteotomy or fracture, and infection with mixed Gram+ and Gram− flora. Statistically significantly higher values of the wound score (that is, lighter wounds) of subjects of the control group in which the wound was definitely covered with a free partial-thickness skin graft compared to the study group can be interpreted as a positive effect of negative pressure therapy.

The average duration of NPWT in our research was 20.69 ± 16.08 days and lasted from 6 to 81 days. Different factors affecting the characteristics of the wound and its healing process, such as the size of the wound, exposed structures in the wound, presence of infection and type of causative agent, comorbidity, and associated injuries, certainly affect the different duration of negative pressure therapy. The application technique of the V.A.C.^®^ therapy system, the type of sponge used, the amount and mode of negative pressure, and possibly the frequency of dressing can also affect the duration of NPWT. A similar conclusion was made by Rasool et al. [22]. 

The number of dressings/changes of Granufoam^®^ sponges was significantly lower in the study group (2.83) compared to the control group (9.00), which is one of the main advantages of negative pressure therapy compared to classic wet to moist dressings. The Granufoam^®^ sponge change interval recommended by the manufacturer of the VAC™ system is 48–72 h. However, that period can vary in practice and be both shorter and longer, due to objective circumstances and the personal preferences of the surgeon. Sponge changes in our patients were planned and carried out two times a week. Dressing the wound on the third or fourth day is much more convenient than daily dressing for patients, parents, and medical staff. Patients perceive dressing as a stressful and unpleasant event and very often do not want to cooperate with the medical staff, which negatively impacts the wound-healing process. This problem is especially pronounced in the pediatric population, where sometimes it is not possible to establish communication with a child who does not understand the procedure they are undergoing and why it is necessary. 

Each dressing leads to the cooling and destruction of the wound microenvironment, hypoxia, and decreasing mobility of leukocytes and phagocytes [23]. It is known that when water evaporates from a surface, the temperature of that surface decreases. In an open wound, where nothing prevents water from evaporating, the tissue temperature can drop to 21 °C. The gauze used to cover the wound does not greatly reduce water evaporation; hence, the measured temperature of these wounds ranges between 25 °C and 27 °C, i.e., 10 °C below normal tissue temperature [23]. WtM dressings prolong the inflammatory phase of wound healing, which has a negative effect on wound healing [24]. Moreover, WtM dressings have a greater potential to expose the patient and hospital staff to possibly infectious contents of the wound, whereas with the V.A.C.^®^, in addition to the lower number of dressings, the material from the wound is stored in a canister, isolated from the environment, so that a minimal amount of harmful content remains in the actual wound. Given that children tolerate V.A.C.^®^ sponge changes better than WtM dressings, the wound can be dressed until an adequate surface is created to cover the defect, whereas when using WtM dressings, there is pressure to close the wound as soon as possible to minimize the number of painful interventions while the granulation tissue is still forming. Therefore, we noted in this research that the period from the injury to the definitive covering of the wound was less in the control group. A significantly higher number of outpatient dressings after discharge from the hospital in the control group also speaks in favor of this approach.

The goal of treating complicated wounds is to provide definitive coverage without the use of complicated reconstructive procedures, especially free or vascularized distant flaps, which are associated with complications and morbidity of the donor site. All wounds in this study were successfully and permanently closed in one of four ways: direct suture, free partial-thickness skin autograft, local skin flap, or spontaneous epithelization, or a combination of these methods. This means that complicated extremity wounds are closed without the use of free or vascularized flaps, meaning simpler methods. There was no statistically significant difference between the study and control groups regarding the method of definitive wound coverage. However, since the patients in the study group had more severe wounds according to the wound score, these data show that even severe wounds can be closed with simple methods when NPWT is used. 

The total duration of antibiotic therapy was statistically significantly longer in the study group compared to the control group (Table 3). Apart from the greater severity of wounds in the study group, this can be further explained by the period of 10 days before the start of NPWT, in which the patients from the study group were treated with the WtM dressing method. 

The duration of analgesic therapy was not significantly different between the study and control groups, both in total duration and in the number of days of analgesic therapy after the start of NPWT administration.

In the available literature, there are no data on the application of blood transfusion and blood derivatives in pediatric patients who were treated with the NPWT method for wounds on the extremities. Only Shilt et al. stated that three patients, who were treated with the traditional method of WtM dressing, due to lower extremity injuries sustained from a lawnmower, received a transfusion of erythrocytes, whereas none of the patients treated with NPWT received a transfusion [5]. Ten patients from the study group in our research received an average of 4.40 ± 2.01 doses, i.e., 1540.00 ± 703.88 mL of erythrocytes, while eight patients from the control group received an average of 1.88 ± 1.35, i.e., 611, 25 ± 509.36 mL of erythrocytes. This difference is statistically significant, both in the number of doses (Z = 2.762; *p* = 0.006) and in the amount of erythrocytes received (Z = 2.734; *p* = 0.006). This can be explained by the fact that patients from the study group had more severe injuries, given that the wound score was statistically significantly lower in the study group, 5.47 ± 1.98, than in the control group, 6.58 ± 1.10 (Z = 2.133; *p* = 0.033). The difference in transfusion of fresh frozen plasma was not statistically significant. 

Baharestani et al. divided the complications that may occur when using the NPWT system in the pediatric population into serious (major) and mild (minor) complications [10]. In the available literature dealing with the treatment of extremity wounds using negative pressure therapy in children, not a single fatal outcome has been noted as a consequence of NPWT.

In two patients from the study group, bleeding occurred due to a serious complication during NPWT. In the first patient, there was profuse bleeding from the granulation tissue immediately after the second Granufoam^®^ sponge change. Negative pressure therapy was therefore discontinued and continued after 7 days, while NPWT of the other two wounds on the same extremity was not discontinued. In another patient with a long-term wound infection due to an open fracture of the proximal metaphysis of the tibia that was treated with external fixation, immediately after changing the Granufoam^®^ sponge, profuse bleeding occurred from the posterior tibial artery, which was not directly exposed in the wound, but its wall was probably damaged by long-term infection. Negative pressure therapy was immediately discontinued, and the vessel was reconstructed by a vascular surgeon. Transfusion was necessary in both patients with serious complications. In our study, there were no complications in terms of wound infections during NPWT. All wound infections were confirmed microbiologically prior to initiation of the V.A.C.^®^ therapy system.

A significantly greater number of mild (minor) complications in the study group had practically no clinical significance because all these complications were related exclusively to a stoppage in the functioning of the V.A.C.^®^ therapy system; hence, they could not have even occurred in the control group and were related to the loss of vacuum due to peeling of the foil, obstruction of the system/sponge with tissue detritus, and overfilling of the canister. All minor complications were immediately resolved by correcting or replacing the system. The ability to quickly respond to the occurrence of a blockage in the V.A.C.^®^ system is one of the advantages of its application in hospital environments compared to home use. In the control group, two patients had serious complications, namely complete lysis of the skin flap due to infection in one patient and severe scar contracture of the ankle joint, with secondary pes equinus, which required subsequent reconstructive surgeries.

Also, in 27 patients in whom treatment was started with the WtM dressing method, it was subsequently switched to NPWT using the V.A.C.^®^ system. Although these patients were included in the study group, the need for switching to NPWT represents a failure of the commenced WtM dressing therapy and can be seen as a kind of complication arising from this method. 

One of the important parameters in evaluating the treatment outcome is the quality of the scar, which was determined in our research using the Observer Scar Assessment Scale (OSAS). There was a statistically significant difference between the OSAS of the study group and the control group, that is, the quality of the scars was significantly better in the study group.

## 5. Conclusions

Based on the results of our research, the conclusion is that negative pressure wound therapy is more effective in the treatment of complicated wounds in children because it requires fewer wound dressings, can be applied in severely complicated wounds, definitive wound coverage is possible with a simpler technique (so-called “downgrading”), a better scar score was achieved, and the number of times of switching from WtM dressing therapy to NPWT was high in our study. The WtM dressing therapy was more effective in terms of shorter hospitalization time. Both methods of therapy are equally safe, considering the same number of serious (major) complications. There was no difference in the effectiveness of NPWT in the treatment of complicated extremity wounds in children between the subgroup of subjects with traumatic wounds and the subgroup of subjects with surgically created wounds. Negative pressure wound therapy has been shown to be equally effective regardless of the mechanism of wound formation.

Based on the evidence presented so far, a clinical recommendation can be made:

“In all complicated wounds, especially those larger than 50 cm^2^, with proven or threatened infection, with soft tissue defect and exposed neuro-vascular elements, bony structures (including fractures and osteotomies) and osteosynthesis, after hemostasis, surgical treatment and debridement wounds, treatment should begin immediately with NPWT using the V.A.C.^®^ system within 24–48 h of injury.”

## Figures and Tables

**Figure 1 children-10-00298-f001:**
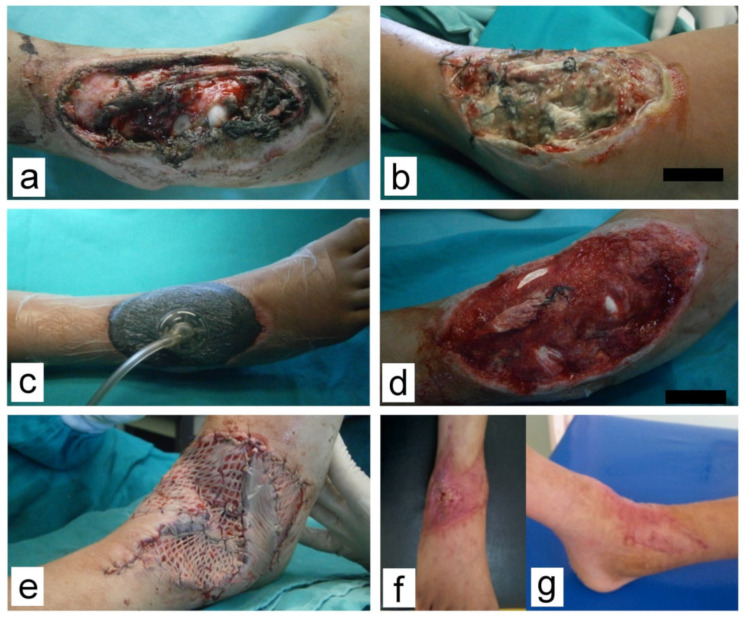
Complex wound with soft tissue defect in the region of the right talocrural joint. (**a**) At presentation; (**b**) Before debridement and V.A.C.^®^, 4 days post injury; (**c**) After V.A.C.^®^ was applied; (**d**) Development of healthy granulations after 7 days of VAC treatment; (**e**) Wound was covered by STSG and local flaps after 13 days; (**f**,**g**) At the one-month follow-up.

**Figure 2 children-10-00298-f002:**
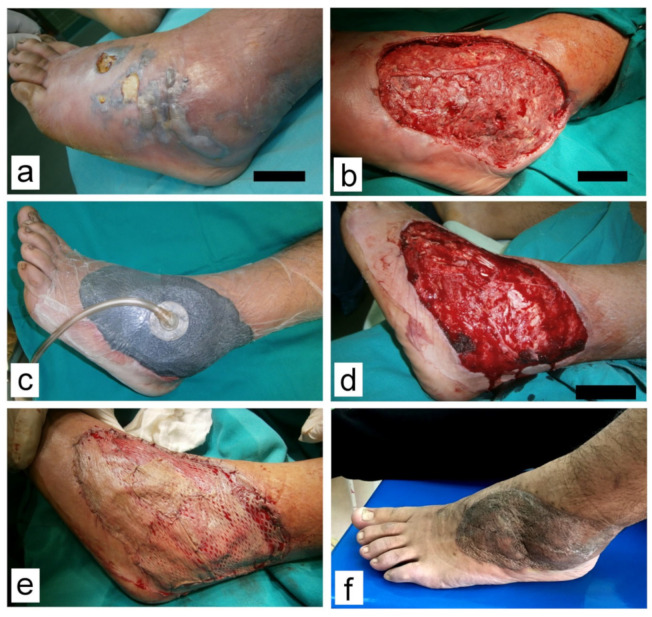
Wound on the right foot with necrotizing fasciitis caused by a streptococcus β-hemolithicus infection. (**a**) Prior to debridement; (**b**) Soft tissue defect after debridement 5 days post injury; (**c**) Marked decrease in soft tissue swelling during V.A.C.^®^ treatment; (**d**) Granulation tissue after 14 days on V.A.C.^®^. Note adhered parts of black foam that had to be surgically removed; (**e**) Wound was covered with STSG after 20 days; (**f**) At the 2-month follow-up visit.

**Figure 3 children-10-00298-f003:**
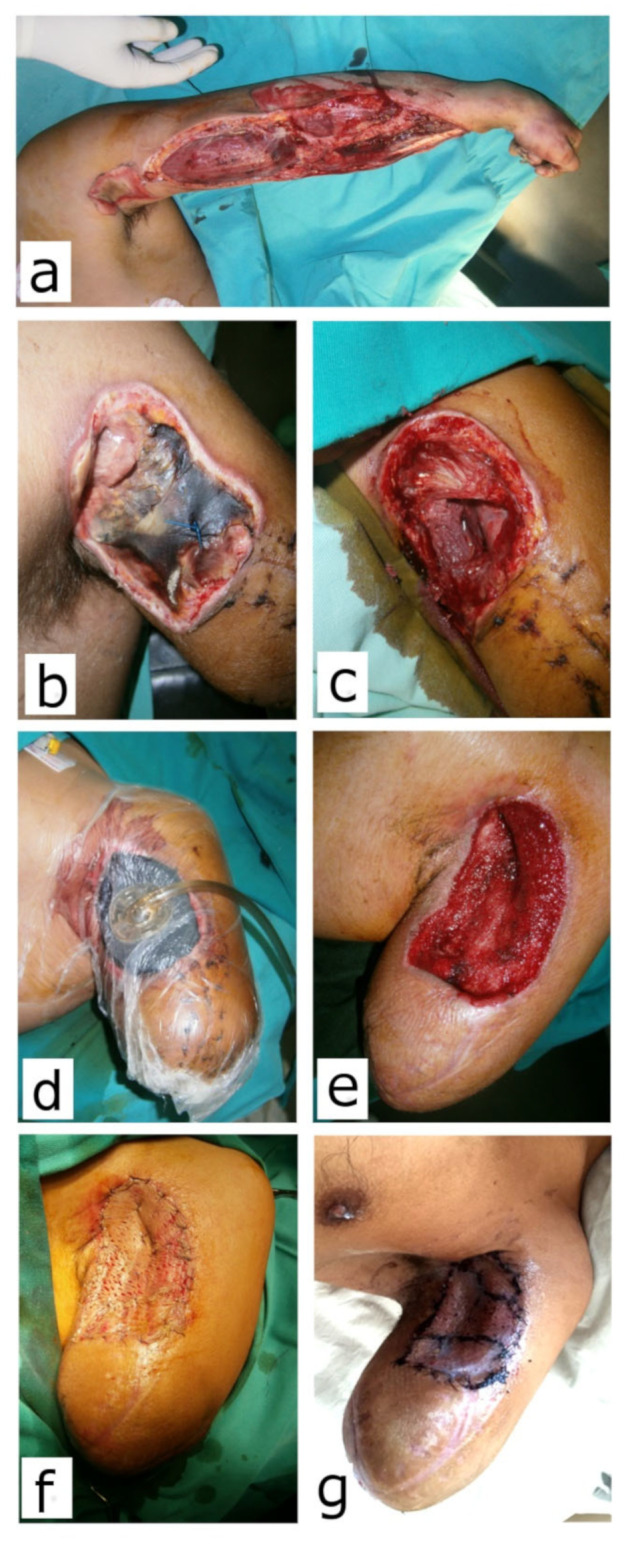
Arm amputation due to electrocution in a 16-year-old boy (Case 6). (**a**) Fasciotomy was performed immediately after injury; (**b**) Area of the coagulation necrosis on the base of the stump; (**c**) After debridement, 24 days post injury; (**d**) V.A.C.^®^ in place; (**e**) Granulation tissue after 3 weeks on V.A.C.^®^; (**f**) Wound covered with STSG; (**g**) Fourteen days after grafting, before referral to prosthetic treatment.

**Figure 4 children-10-00298-f004:**
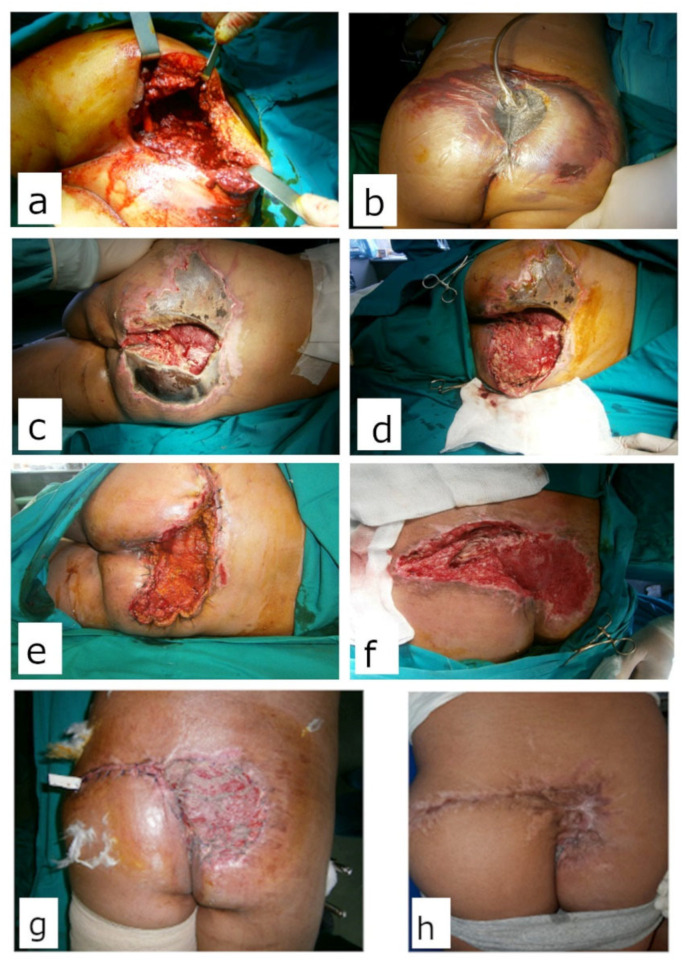
Twelve-year-old boy, hit by a train. (**a**) During primary debridement. Large soft tissue defect with decollement and exposed sacrum (**b**) Before first V.A.C.^®^ dressing change; (**c**) Large soft tissue necrosis during NPWT treatment. Mixed Gram+/− flora was isolated; (**d**) Excisional debridement; (**e**) Definitive coverage of the wound on the left gluteus with direct sutures. Healthy granulations on the right gluteus; (**f**) Before definitive coverage of the right gluteus 56 days post injury and after 16 dressing changes; (**g**) Wound was covered with meshed STSG; (**h**) Three months after coverage.

**Table 1 children-10-00298-t001:** Patients demographics and wound characteristics.

	NPWT Group (*n* = 34)	WtM Group (*n* = 24)	Significance *p*=
Sex			0.046
Male n (%)	29 (85.3)	15 (62.5)	
Female n (%)	5 (14.7)	9 (37,5)	
Age	14.58 ± 2.85	10.83 ± 4.41	<0.001
Type of wound			0.090
Traumatic n (%)	15 (44.1)	16 (66.7)	
Surgical excisional n (%)	19 (55.9)	8 (33.3)	
Wound area	83.17 ± 65.97 cm^2^	58.38 ± 58.12 cm^2^	0.072
Exposed structures			
Bone	17 (50.0)	15 (62.5)	0.346
Tendons	9 (26.5)	12 (50.0)	0.066
Neurovascular structures	11 (32.4)	6 (25.0)	0.545
Fracture or osteotomy	11 (33.3)	9 (37.5)	0.745
Osteosynthesis	8 (23.5)	1 (4.2)	0.045
Microbiology isolate			0.020
Gram +	3 (8.8)	7 (29.2)	
Gram−	4 (11.8)	2 (8.3)	
Mixed (Gram+ and Gram−)	14(42.1)	2 (8.3)	0.006
Wound score	5.47 ± 1.98	6.58 ± 1.10	0.033

**Table 2 children-10-00298-t002:** Etiology of traumatic and surgical wounds.

	NPWT Group *n* (%)	WtM Group *n* (%)
Traumatic wounds		
Traffic accidents	9 (24.47)	11 (45.83)
Injuries with agricultural machinery	4 (11.76)	1 (4.17)
Electrocution	1 (2.94)	0
Fall of a heavy object	1 (2.94)	0
Injuries by a sharp or pointed object	0	4 (16.67)
Total	15 (44.12)	16 (66.67)
Surgical excisional wounds		
Excision of pilonidal sinus	6 (17.64)	0
Excision/debridement of a decubital wound	4 (11.74)	0
Deep soft tissue infection	6 (17.64)	7 (29.17)
Soft tissue and bone infection	3 (8.82)	1 (4.17)
Total	19 (55.82)	8 (33.34)

**Table 3 children-10-00298-t003:** Treatment overview.

	NPWT Group (Mean ± SD)	WtM Group (Mean ± SD)	Significance *p*=
Duration of hospitalization (days)	36.44 ± 26.73	19.46 ± 16.31	*p* = 0.002
Time from injury to beginning of NPWT or WtM treatment (days)	10.27 ± 11.11	4.54 ± 8.46	0.023
Time from beginning of treatment to definitive wound closure (days)	19.76 ± 16.16	12.20 ± 11.58	*p* = 0.023
Number of dressing changes	2.83 ± 5.25	9.00 ± 7.24	0.009
Number of dressing changes in the outpatient clinic (after discharge)	0.97 ± 1.33	2.54 ± 2.34	0.001
Scar score (OSAS)	10.38 ± 4.29	13.00 ± 4.53	0.0434
Fever (days)	2.71 ± 5.33	4.04 ± 5.82	*p* = 0.233
Antibiotic therapy (days)	19.39 ± 15.12	14.50 ± 9.35	0.251
Analgetic therapy (days)	7.97 ± 12.25	6.58 ± 6.81	0.464
Transfusion of erythrocytes (ml)	1540.00 ± 703.88	611.25 ± 509.36	0.006
Transfusion of fresh frozen plasma (ml)	1077.50 ± 634.48	605.00 ± 210.63	0.069
Complications	*n* (%)	*n* (%)	
Minor	8 (23.5)	0 (0.0)	
Major	2 (5.9)	2 (8.3)	

**Table 4 children-10-00298-t004:** Modality of definitive wound closure.

	NPWT Group *n* (%)	WtM Group *n* (%)
Direct suture	10 (29.4)	10 (41.7)
Split thickness skin graft (STSG)	12 (35.3)	6 (25.0)
Local flap	1 (2.9)	0 (0.0)
Spontaneous epithelization	4 (11.8)	1 (4.2)
Combination of two methods	7 (20.6)	7 (29.2)

## Data Availability

The data presented in this study are available on request from the corresponding author.

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
