# Peer review of "Comparison of Negative Pressure Wound Therapy (NPWT) and Classical Wet to Moist Dressing (WtM) in the Treatment of Complicated Extremity Wounds in Children"

_children, 2023, doi:10.3390/children10020298_

Round 1

Reviewer 1 Report

First of all, abbreviations should be presented and after that used uniformly according to the policy of the journal.

There is no clear definition of a complicated wound although you can deduct it from the first sentence in Introduction.

This sentence should be revised adequately.

he effects of negative pressure at both macroscopic and microscopic levels facilitate the formation of healthy granulation tissue [7], promoting spontaneous wound closure or coverage (clousure) with direct suture, free split-thickness skin grafts (STSG), local flaps or secondary intention [14].

The system was set to a continuous negative pressure of -100 to -125mm Hg according to the patient’s age.

You had patients of 1 year also and -100 is considered rather high. Is there any data to support your decision to choose so high values of NP.

Statistical analysis was performed by a specialist in medical statistics.

Please define which methods and with which application.

Please define the type of the study. Was it conducted in single institution or.....

The significance of our study is that it is the first prospective research with 213a statistically significant number of patients that compares NPWT and the traditional wet 214to moist wound dressing.

how can it be prospective when the ethical committee approved it in 2018 and patients were enrolled already in 2014?

Part of table 1 is out of the center - needs correction

Was petroleum jelly used or vaseline gauze? While vaseline gauze is considered one of the protection barriers when NPWT is applied you should clarify why you used petroleum jelly and maybe discuss of its use by other authors.

Although infection is cited as one of the most significant com-263plications of NPWT, the results of this study show that the application of negative pressure 264effectively suppresses infection in the treatment of complicated wounds.

Can you explain this statement with numbers: decrease in number of infected wounds if you took swabs during changes.

Other authors have similar experiences. In the study by McCord and colleagues (of 26682 wounds, 26 had clear signs of infection before the start of NPWT, which was confirmed 267by wound swab and blood culture [22].

and how many had the infection after NPWT. This is important to support your results.??

Ten patients from the control group in our 345study received an average of 4.40±2.01 doses, i.e., 1540.00±703.88 ml of erythrocytes, while 3468 patients from the control group received an average of 1.88±1.35, i.e., 611, 25±509.36 ml 347of erythrocytes.

Probably a mistake - twice a control group???

Also, in 27 patients in whom treatment was started with the WtM dressing method, 382it was subsequently switched to NPWT using the V.A.C.® system. Although these patients 383were included in the study group, the need for switching to NPWT represents a failure of 384the commenced WtM dressing therapy and can be seen as a kind of complication arising 385from this method.

Does that mean that NPWT was used as a first line treatment in only 7 patients? Is then prudent to say that WtM failed or was it more a matter of concept like in Furnier where most researchers start with WtM and then after several (2-4 days) change to NPWT.

Author Response

Dear reviewer, first of all we would like to thank you for setting aside your valuable time to evaluate this paper and giving us insight into flaws that we were not aware of. Your comments are valuable to us and appreciated. We have made changes to our manuscript which you can evaluate, and here we present our answers to your comments:

  1. First of all, abbreviations should be presented and after that used uniformly according to the policy of the journal. Noted and corrected.

  1. There is no clear definition of a complicated wound although you can deduct it from the first sentence in Introduction - Noted, we’ve added a referenced definition in the introduction.

  1. The effects of negative pressure at both macroscopic and microscopic levels facilitate the formation of healthy granulation tissue [7], promoting spontaneous wound closure orcoverage (clousure) with direct suture, free split-thickness skin grafts (STSG), local flaps or secondary intention [14].This sentence should be revised adequately. Sentence corrected.

  1. The system was set to a continuous negative pressure of -100 to -125mm Hg according to the patient’s age. You had patients of 1 year also and -100 is considered rather high. Is there any data to support your decision to choose so high values of NP.Actually, the patient that was one year old was treated by traditional wet to moist dressing changes. The youngest patient treated by NPWT was 6 years old. The 1-18 was the range for the entire group of patients, we’ve now added a range for each group separately.

  1. Statistical analysis was performed by a specialist in medical statistics. Please define which methods and with which application. Added.

  1. Please define the type of the study. Was it conducted in single institution or..... – The patients are from a single institution, however we consulted colleagues from other institutions who have more previous expirience in using the VAC system. So this is a single institution prospective study.

  1. The significance of our study is that it is the first prospective research with a statistically significant number of patients that compares NPWT and the traditional wet to moist wound dressing. how can it be prospective when the ethical committee approved it in 2018 and patients were enrolled already in 2014? Unfortunately there has been a mistake during the writing process, namely the research started from February 2018 after the ethic committee aprroval. Thank you for pointing out this mistake.

  1. Part of table 1 is out of the center - needs correction – Corrected.

  1. Was petroleum jelly used or vaseline gauze? While vaseline gauze is considered one of the protection barriers when NPWT is applied you should clarify why you used petroleum jelly and maybe discuss of its use by other authors. This was a mistranslation, actually vaseline gauze was used, corrected.

  1. Although infection is cited as one of the most significant complications of NPWT, the results of this study show that the application of negative pressure effectively suppresses infection in the treatment of complicated wounds.Can you explain this statement with numbers: decrease in number of infected wounds if you took swabs during changes. We took swabs during dressing changes mainly to correct antibiotic treatment when needed, but we relied mostly on clinical appereance of the wound to estimate if the wound was infected or not. Therefore NPWT was effective in the sense that it reduced exrections, swelling, redness, promoted granulation tissue, reduced the size of the wound etc. Sometimes the wounds were clinicaly wiithout signs of infections but swabs were positive for bacteria which indicated colonisation but not infections, which is why we didn’t include swabs in the final results.

  1. Other authors have similar experiences. In the study by McCord and colleagues (of 26682 wounds, 26 had clear signs of infection before the start of NPWT, which was confirmed by wound swab and blood culture [22]. And how many had the infection after NPWT. This is important to support your results.?? We’ve found that data and put it in the paper.

  1. Ten patients from the control group in our study received an average of 4.40±2.01 doses, i.e., 1540.00±703.88 ml of erythrocytes, while 8 patients from the control group received an average of 1.88±1.35, i.e., 611, 25±509.36 ml of erythrocytes.Probably a mistake - twice a control group??? Yes, it is a mistake, the first group is the study group.

  1. Also, in 27 patients in whom treatment was started with the WtM dressing method, it was subsequently switched to NPWT using the V.A.C.® system. Although these patients were included in the study group, the need for switching to NPWT represents a failure of the commenced WtM dressing therapy and can be seen as a kind of complication arising from this method. Does that mean that NPWT was used as a first line treatment in only 7 patients? Is then prudent to say that WtM failed or was it more a matter of concept like in Furnier where most researchers start with WtM and then after several (2-4 days) change to NPWT. Actually we believe that this is a failure of the WtM because we didn’t switch from WtM to NPWT intentionaly. We had to switch to NPWT because WtM was ineffective in controling the healing of the wound.

Reviewer 2 Report

Authors present herein an observational study that tends to position itself as a confirmatory study and is mainly motivated by limited knowledge with regards to the indications and outcomes of negative pressure wound therapy in complicated wounds of the extremities in children I have several comments on this study:
-the general design of this study is not reported in the materials and methods section, where it should be standard; it should ideally be described in the methods section and have a reflection both in the title ad the abstract
-going through the manuscript, the reader's impression is that of a retrospective study (especially given the statements about patient selection in the methods section and the fact that groups were created by dividing causes of death based on treatment and the primary wound type. Then in the discussion section the study is labeled as a prospective one. However, there is no mention to an a priori protocol, to researchers' criteria for group assignment, exclusion criteria, etc.
-(important, particularly so in studies examining children) ethical review details must be produced, alongside informed consent of parents or guardians
-throughout the manuscript, there is no definition of what is a complicated wound in the context of this study. Several definitions exist in the literature, at times distinguished by nuances, but at least one definition must be given or referenced, and complicated wounds are the main subject of this study
-likewise, outcomes should be defined alongside their units of measure
-statistical analysis: there is no indication of the types of variables that were considered, and no indication of the types of tests that were run (eg Chi-square, Student's T, etc). Without such indication, statistical significance loses its sense because it is not referred to a means of comparison.

Author Response

Dear reviewer, first of all we would like to thank you for setting aside your valuable time to evaluate this paper and giving us insight into flaws that we were not aware of. Your comments are valuable to us and appreciated. We have made changes to our manuscript which you can evaluate, and here we present our answers to your comments:

-the general design of this study is not reported in the materials and methods section, where it should be standard; it should ideally be described in the methods section and have a reflection both in the title ad the abstract . Noted, we’ve added the study design in the “materials and methods” section.

-going through the manuscript, the reader's impression is that of a retrospective study (especially given the statements about patient selection in the methods section and the fact that groups were created by dividing causes of death based on treatment and the primary wound type. Then in the discussion section the study is labeled as a prospective one. However, there is no mention to an a priori protocol, to researchers' criteria for group assignment, exclusion criteria, etc. The group assigment was based on the preference and experience of the treating physician. Also, we clarified inclusion and exclusion critertia in
"Materials and methods" section.

-(important, particularly so in studies examining children) ethical review details must be produced, alongside informed consent of parents or guardians. We’ve included the date and protocol code of  the Ethics Committee descision in the original manuscript. We’ve now also added that decision in the „materials and methods“ section. 

-throughout the manuscript, there is no definition of what is a complicated wound in the context of this study. Several definitions exist in the literature, at times distinguished by nuances, but at least one definition must be given or referenced, and complicated wounds are the main subject of this study. Noted, we’ve added a referenced definition in the introduction.

-likewise, outcomes should be defined alongside their units of measure. We are not exactly sure about this comment. Do you think that we should add units of measure to some of the results? We noticed that there were some units of measures missing in a couple of places and added them, if this was the issue.

-statistical analysis: there is no indication of the types of variables that were considered, and no indication of the types of tests that were run (eg Chi-square, Student's T, etc). Without such indication, statistical significance loses its sense because it is not referred to a means of comparison. Noted, we’ve added the methods used in the “materials and methods” section.

Reviewer 3 Report

The paper is dealing with an interesting and difficult topic. Complicated wound management is especially in children difficult to conduct. Nevertheless a long standing wound management can be a burden for all person dealing with the problem, patients parents, nurse and medical staff. Usually a long period is required to take care of the problem, complicated by difficult interaction between all players. 

The vacuum treatment is certainly an important issue and this study provides a fairly big number.  Even if many institutions are involved and as usual it is also here difficult to follow a unique study protocol.

The results are interesting and as expected from larger adult series. The discussion and conclusions are reasonable. 

The paper is well written and can be accepted with minor corrections. A remark on the role of interactive treatment, and decision making for vacuum therapy in agreement with hospital, outpatient care and territory doctors should be added.  

Author Response

Dear reviewer, first of all we would like to thank you for setting aside your valuable time to evaluate this paper. Your comments are valuable to us and appreciated. We have made changes to our manuscript which you can evaluate. We also agree with your comment that vacuum therapy should be done in agreement with hospital, outpatient care and territory doctors. However at this stage of our research, we have only used the NPWT system on patients who were hospitalised. We don't have any experience in the outpatients setting and therefore we don't feel like we can make any comments in this area yet. However, there are plans for further research to encompass outpatient treatment. 

Round 2

Reviewer 2 Report

The paper address an interesting topic and collects data from a discrete cohort of patients. The study design perhaps is not strong enough to establish whether NPWT is superior over WtM in the treatment of these kind of lesions but for sure adds further useful data on the argument.